# Preparation of Ag_3_PO_4_/TiO_2_(B) Heterojunction Nanobelt with Extended Light Response and Enhanced Photocatalytic Performance

**DOI:** 10.3390/molecules26226987

**Published:** 2021-11-19

**Authors:** Yong Li, Yanfang Liu, Mingqing Zhang, Qianyu Zhou, Xin Li, Tianlu Chen, Shifeng Wang

**Affiliations:** 1Department of Physics, and Innovation Center of Materials for Energy and Environment Technologies, College of Science, Tibet University, Lhasa 850000, China; xzuliyong@utibet.edu.cn (Y.L.); liuyanfang@utibet.edu.cn (Y.L.); mingqing@utibet.edu.cn (M.Z.); Zhouqianyu@utibet.edu.cn (Q.Z.); lixin@utibet.edu.cn (X.L.); 2Institute of Oxygen Supply, Center of Tibetan Studies (Everest Research Institute), Tibet University, Lhasa 850000, China; 3Key Laboratory of Cosmic Rays (Tibet University), Ministry of Education, Lhasa 850000, China

**Keywords:** photocatalyst, heterojunction, photocatalytic degradation, TiO_2_(B) nanobelts

## Abstract

Photocatalytic degradation, as an emerging method to control environmental pollution, is considered one of the most promising environmental purification technologies. As Tibet is a region with some of the strongest solar radiation in China and even in the world, it is extremely rich in solar energy resources, which is ideal for applying photocatalytic technology to its ecological environment protection and governance. In this study, Na_2_Ti_3_O_7_ nanobelts were prepared via a hydrothermal method and converted to TiO_2_∙xH_2_O ion exchange, which was followed by high-temperature calcination to prepare TiO_2_(B) nanobelts (“B” in TiO_2_(B) means “Bronze phase”). A simple in situ method was used to generate Ag_3_PO_4_ particles on the surface of the TiO_2_ nanobelts to construct a Ag_3_PO_4_/TiO_2_(B) heterojunction composite photocatalyst. By generating Ag_3_PO_4_ nanoparticles on the surface of the TiO_2_(B) nanobelts to construct heterojunctions, the light absorption range of the photocatalyst was successfully extended from UV (ultraviolet) to the visible region. Furthermore, the recombination of photogenerated electron–hole pairs in the catalyst was inhibited by the construction of the heterojunctions, thus greatly enhancing its light quantum efficiency. Therefore, the prepared Ag_3_PO_4_/TiO_2_(B) heterojunction composite photocatalyst greatly outperformed the TiO_2_(B) nanobelt in terms of photocatalytic degradation.

## 1. Introduction

The Qinghai–Tibet Plateau is the ecological security barrier of China and the origin of many rivers in China and South Asia. Therefore, it is of great significance to protect and manage its ecological environment [1,2,3]. As an emerging environmental pollution control technology, photocatalytic degradation is considered one of the most promising environmental purification technologies at present because of its advantages of mild reaction conditions, low secondary pollution, sustainability, and environmental friendliness [4,5,6,7]. Since Tibet is one of the regions with the strongest solar radiation in China and even the world, the abundant energy resource is ideal for applying photocatalytic technology to ecological environment protection and governance in Tibet [8,9,10]. TiO_2_ is recognized as one of the best photocatalyst materials with stable properties, low cost, ease of access, non-toxicity and safety, and good resistance to photocorrosion. However, TiO_2_ has a band gap as large as 3.2 eV and hence is photocatalytically active only under UV light, which accounts for merely approximately 5% of solar radiation energy. Therefore, TiO_2_ as a photocatalyst has a low utilization of solar radiation energy and insignificant photocatalytic performance [11,12,13]. However, combining TiO_2_ with other narrow-band-gap semiconductor materials to construct heterojunctions can effectively enhance the photocatalytic performance of TiO_2_. This is because, first, the introduction of narrow-band-gap semiconductors can extend the light-absorption range of TiO_2_ from the UV region to the visible region, greatly increasing the material’s absorption range of the solar spectrum; second, the construction of heterojunctions can effectively promote the separation of photogenerated electrons and holes, thereby improving the quantum yield of photogenerated carriers participating in photocatalytic redox reactions [13,14,15]. Current studies show that the use of Ag_3_PO_4_ compounded with TiO_2_ can effectively enhance TiO_2_ photocatalytic performance, but most studies use common anatase phase TiO_2_ [16,17,18,19], and monoclinic (space group C2/m) TiO_2_(B) has rarely been studied, while TiO_2_(B) has better electrical conductivity, a sparse porous structure, and stronger photogenerated hole oxidation, and these properties make it possible for TiO_2_(B) to obtain stronger photocatalytic performance by compounding [20,21,22,23]. In this study, Na_2_Ti_3_O_7_ nanobelts were successfully prepared via hydrothermal methods and further used as a precursor for ion exchange with H^+^ in 0.1 M dilute HCl to obtain TiO_2_ hydrate, which was dried and calcined to obtain TiO_2_(B) nanobelts. Throughout the preparation process, the morphology of the Na_2_Ti_3_O_7_ nanobelts was maintained. Na_2_Ti_3_O_7_ had good adsorption properties, as did TiO_2_(B) with the same morphology, which was beneficial to its absorption of sunlight and photocatalytic degradation performance. Finally, Ag_3_PO_4_ nanoparticles were generated on the surface of the TiO_2_(B) nanobelts via a simple in situ method, thereby successfully constructing Ag_3_PO_4_/TiO_2_(B) heterojunctions, which substantially enhanced the photocatalytic degradation performance of TiO_2_(B).

## 2. Results and Discussion

Figure 1a,b present SEM (Scanning Electron Microscope) images of Na_2_Ti_3_O_7_ and the TiO_2_(B) nanobelt, respectively, which show that they had the same morphology and were both belt-like. Therefore, TiO_2_(B) prepared from Na_2_Ti_3_O_7_ using the experimental method in this study was able to well-maintain the morphology of Na_2_Ti_3_O_7_. Figure 1c shows EDS (Energy-Dispersive X-Ray Spectroscopy Spectra) of Na_2_Ti_3_O_7_; Na, O, and Ti appear in the spectra, which verifies the chemical composition of the prepared product. Figure 1d shows EDS spectra of TiO_2_(B); O and Ti appear in the spectra, which verifies the chemical composition of the prepared product. Figure 1e shows XRD (X-Ray Diffraction) patterns of TiO_2_, the analysis of which revealed that the TiO_2_(B) nanobelts prepared in this experiment did not belong to the rutile, anatase, or brookite phases common to TiO_2_ in nature, but they did belong to a monoclinic crystal system (space group C2/m).

Figure 2a shows TEM (Transmission Electron Microscopy) diffraction patterns of the TiO_2_(B) nanobelts, and Figure 2b shows the diffraction spots of crystal planes (0 2 0), (0 $\overset{–}{2}$ 0), (6 0 $\overset{–}{3}$), and ($\overset{–}{6}$
0 3) as calculated using the crystal structure data measured via XRD. Comparing the two images revealed that the measured and theoretically calculated results were in good agreement. In addition, the regular lattice pattern of the diffraction spots also indicated a high crystallinity of the prepared TiO_2_ nanobelts. This is also confirmed by Figure 2d, which clearly shows a uniform distribution of TiO_2_ grains. Figure 2c shows that the (0 2 0) crystal planes were uniformly arranged with a measured crystal plane spacing of 0.188 nm, which agreed very well with that measured via XRD. The above TEM test analysis further confirmed that the crystal structure of the TiO_2_(B) nanobelts did not belong to the rutile, anatase, or brookite phases of TiO_2_ that are common in nature but was a new phase of TiO_2_ that is not commonly found in nature.

Figure 3 shows XRD patterns of the samples. The phases of the TiO_2_(B) nanobelts and prepared Ag_3_PO_4_ are analyzed in the left panel of Figure 3, with the standard powder diffraction file (PDF) cards PDF#06-0505 and PDF#46-1237, corresponding to Ag_3_PO_4_ and TiO_2_(B), respectively. It can be clearly seen from the diffraction patterns that the crystallinity of Ag_3_PO_4_ was significantly better than that of the TiO_2_(B) nanobelts, and the diffraction peaks of Ag_3_PO_4_ corresponded very well to the standard PDF card. The right panel of Figure 3 shows XRD diffraction patterns of the composites of Ag_3_PO_4_ and TiO_2_(B) with different molar ratios. There were very obvious diffraction peaks of Ag_3_PO_4_ in the composites, and the relative intensity of the peaks did not change significantly with the increase in the molar ratio of Ag_3_PO_4_. Among the diffraction peaks of TiO_2_(B), only the strongest diffraction peak, corresponding to the (110) plane, was weakly reflected in the composite, and it weakened with the increase in the molar ratio of Ag_3_PO_4_.

Figure 4 shows SEM and mapping images of the Ag_3_PO_4_/TiO_2_(B) composite, where Figure 4a shows a SEM image when the molar ratio of Ag_3_PO_4_ to TiO_2_ was 0.4:1, and Figure 4b shows the SEM image when the molar ratio increased to 1.5:1. The Ag_3_PO_4_ particles were clearly attached to the surface of belt-like TiO_2_(B), and the number of Ag_3_PO_4_ particles increased as the molar ratio of Ag_3_PO_4_ increased. EDS mapping (Figure 4c–f) showed that the sample contained Ti, O, Ag, and P elements, which were uniformly distributed in the area where the composite was located, indicating that the composite contained Ag_3_PO_4_ and TiO_2_(B), and the two were distributed relatively uniformly.

The left panel of Figure 5 shows the photocatalytic degradation curves of RhB solution by Ag_3_PO_4_/TiO_2_(B) composites with different molar ratios. It can be seen that the TiO_2_(B) nanobelts degraded RhB slowly, and the degradation rate was less than 20% after 30 min of degradation. After combining with Ag_3_PO_4_, the photocatalytic degradation of RhB by the composite product was improved, particularly when increasing the molar ratio of Ag_3_PO_4_. When the molar ratio of Ag_3_PO_4_ to TiO_2_(B) increased to 1.5:1, within 30 min, the composite photocatalyst achieved nearly 100% degradation of RhB, and the photocatalytic performance increased by nearly five times, which was much higher than the photocatalytic degradation performance of commercial P_25_. The right panel of Figure 5 shows actual photographs of the color change of RhB solution due to photocatalytic degradation by Ag_3_PO_4_/TiO_2_(B) composites with different molar ratios. These photographs visually demonstrate the rapid enhancement of the photocatalytic performance of the composite with the increase in the molar ratio of Ag_3_PO_4_. The Ag_3_PO_4_/TiO_2_(B) composite was obtained on the surface of the TiO_2_(B) nanobelts through the in situ growth of Ag_3_PO_4_ crystal grains. Compared with that of the TiO_2_(B) nanobelts, the photocatalytic performance of the composite increased rapidly with the increase in the molar ratio of Ag_3_PO_4_. When the molar ratio of Ag_3_PO_4_ to TiO_2_(B) reached 1.5:1, the photocatalytic performance of the composite far exceeded (was nearly five times) that of the pure TiO_2_(B) nanobelt. We believe that the enhancement of the photocatalytic ability of the composite was, first, due to the introduction of Ag_3_PO_4_, which greatly expanded the absorption range of the material within the solar spectrum and enhanced the absorption of visible light [24,25,26] and, second, because of the construction of heterojunctions, which promoted the separation of photogenerated electron–hole pairs and inhibited their recombination [27,28,29]. In addition, it can be seen from the mapping results that the two substances constructed heterojunctions that were highly uniformly distributed, which gave full play to their role in the composite.

Figure 6 shows the absorbance curves of the samples against a continuous spectrum. This test investigated the variation curves of absorbance of the TiO_2_(B) nanobelts and Ag_3_PO_4_ in the visible region (wavelength > 400 nm). It was obvious that the light-absorption ability of Ag_3_PO_4_ was significantly higher than that of the TiO_2_(B) nanobelts in the visible region. After the TiO_2_(B) nanobelts and Ag_3_PO_4_ formed a composite, its visible-light-absorption ability was enhanced to some extent compared with that of the TiO_2_(B) nanobelts, indicating that the introduction of Ag_3_PO_4_ enhanced the visible light absorption ability of the composite.

To investigate the recombination and migration of photogenerated electrons before and after the combination of Ag_3_PO_4_ and TiO_2_(B), we tested their PL (photoluminescence) spectra and photocurrent curves, respectively, as shown in Figure 7. As can be clearly seen from the PL spectrum in the left panel of Figure 7, the peaks of the PL lines of the TiO_2_(B) nanobelts were the strongest, while the peaks of the PL lines of the composite decreased rapidly with the increase in the molar ratio of the added Ag_3_PO_4_. Stronger fluorescence reflects an easier recombination of the photogenerated electron–hole pairs of the substance, making it difficult for the photogenerated electron–hole pairs to migrate to the surface of the catalyst to participate in the photocatalytic redox reaction, which is not conducive to the performance of photocatalysis [30,31,32]. The right panel of Figure 7 shows the photocurrent curves of the samples. The photocurrents of pure TiO_2_(B) nanobelts and Ag_3_PO_4_ nanoparticles were the lowest. In comparison, the photocurrent of Ag_3_PO_4_/TiO_2_(B) composite was enhanced and increased continuously with the increase in the molar ratio of Ag_3_PO_4_, reaching a maximum when the molar ratio of Ag_3_PO_4_ was increased to 1.5:1. These two sets of curves well explained the underlying reason for the variation pattern of photocatalytic performance in Figure 5 and also indirectly proved that Ag_3_PO_4_ nanoparticles grown on the surface of TiO_2_(B) nanobelts through our simple in situ method indeed successfully constructed a Ag_3_PO_4_/TiO_2_(B) heterojunction composite, effectively promoting the separation of photogenerated charges and thereby enhancing the photocatalytic performance of the composite catalyst.

## 3. Materials and Experiment

### 3.1. Materials

Regarding the chemicals used in the experiment, NaOH and nano-TiO_2_ were purchased from Aladdin Industries, Inc.(Shanghai, China), Na_2_HPO_4_∙12H_2_O from Chengdu Jinshan Chemical Reagent Co., Ltd. (Chengdu, China), AgNO_3_ from Sinopharm Chemical Reagent Co., Ltd. (Shanghai, China), and Rhodamine B (RhB, C_28_H_31_CIN_2_O_3_) from Beijing Solarbio Technology Co., Ltd. (Beijing, China). All the reagents were analytically pure and used without further purification.

### 3.2. Preparation of TiO_2_(B) Nanobelt, Ag_3_PO_4_, and Ag_3_PO_4_/TiO_2_(B)

Nano-TiO_2_ (1 g) was added to 100 mL of 10 M NaOH aqueous solution, stirred evenly, poured into a 150 mL hydrothermal reactor, heated to 200 °C, and held for 24 h before cooling naturally. Then, the reaction product was taken out of the reactor and rinsed with a large amount of deionized (DI) water to a nearly neutral pH, which was followed by vacuum filtration and drying to obtain the Na_2_Ti_3_O_7_ nanobelts. After that, Na_2_Ti_3_O_7_ was soaked in 0.1 M dilute HCl for 72 h to allow ion exchange between H^+^ and Na^+^ in Na_2_Ti_3_O_7_ to obtain TiO_2_∙xH_2_O, which was rinsed with a large amount of DI water until the pH of the solution was nearly neutral, vacuum filtered, dried at 60 °C in a blast drying oven, calcined at 500 °C in a muffle furnace with a heating rate of 1 °C/min for 2 h, and cooled naturally to obtain the TiO_2_(B) nanobelts. To prepare Ag_3_PO_4_/TiO_2_(B), 100 mg of the prepared TiO_2_(B) nanobelts was weighed first, and corresponding masses of Ag_3_PO_4_ were calculated according to molar ratios of (1:0.4), (1:0.8), and (1:1.5), respectively, as were the corresponding masses of AgNO_3_ and Na_2_HPO_4_∙12H_2_O required for the preparation of Ag_3_PO_4_. Then, 100 mg of the TiO_2_ nanobelts and the corresponding mass of Na_2_HPO_4_∙12H_2_O were added into 100 mL of DI water, which was followed by ultrasonication for 30 min. Next, the corresponding mass of AgNO_3_ was dissolved in 50 mL of DI water and then slowly added into the sonicated mixture with constant stirring. Finally, the mixture was vacuum-filtrated and dried to obtain composites with different molar ratios, which were labeled as Ag_3_PO_4_/TiO_2_(B) (0.4:1), Ag_3_PO_4_/TiO_2_(B) (0.8:1), and Ag_3_PO_4_/TiO_2_(B) (1.5:1). Ag_3_PO_4_ was directly obtained by mixing a certain proportion of AgNO_3_ and Na_2_HPO_4_∙12H_2_O solution, which was followed by vacuum filtration and drying.

### 3.3. Analysis and Testing

Field emission scanning electron microscopy (FE-SEM, Gemini SEM 300, Manufactured by Zeiss, Oberkochen, Germany) and field emission transmission electron microscopy (FE-TEM, FEI Talos F200X, Manufactured by FEI, Hillsboro, OR, USA) were used to observe and analyze the morphology of the samples as well as test and analyze the elemental composition and distribution of the samples. FE-TEM (FEI Talos F200X, Manufactured by FEI, USA) and X-ray powder diffractometry (XRD, Bruker D8 Advance, Manufactured by Bruker, Bremen, Germany) were used to test and analyze the crystal structure and phase of the samples. A fluorescence spectrometer (PL FLS 1000/FS5, Manufactured by Edinburgh, UK) was used to test the fluorescence emission spectra of the samples. An electrochemical workstation (CHI-760E, Manufactured by Chenhua, Shanghai, China) was used to test the photocurrent of the samples. A UV-Vis spectrophotometer (UV-1200, Manufactured by Macy, Shanghai, China) was used to test and analyze changes in photocatalytic degradation performance, as well as obtain absorbance curves of the samples for RhB. In addition, a muffle furnace (KSL-1700X, Manufactured by Kejing, Hefei, China), a blast drying oven (DHG-9246A, Manufactured by Kejing, Hefei, China), and a benchtop high-speed centrifuge (LC-LX-H185C, Manufactured by Lichen, Shanghai, China) were also used to prepare the experimental materials.

### 3.4. Photocatalytic Performance Test

Each sample (20 mg) was added to 100 mL of 20 mg/L RhB solution and sonicated for 30 min. The mixture was placed under a light source simulated by a solar simulator (Solar-500Q, Manufactured by Newbit, Beijing, China), with the intensity of the light source adjusted to obtain a light intensity of 600 W/m^2^ at the liquid surface. Approximately 6 mL of the experimental liquid was taken every 10 min and placed into the centrifuge for 10 min at 10,000 r/min. The supernatant was taken and tested by the UV-Vis spectrophotometer for absorbance A_n_ (n = experimental serial number, with 1 for the first sample taken). Since the absorbance is proportional to the concentration of the solution (C_n_), then there is a solution concentration ratio C_n_/C_0_ = A_n_/A_0_ (A_0_ = absorbance of the 20 mg/L of RhB solution, C_0_ = concentration of the 20 mg/L of RhB solution). The absorbance of each sample was measured at 10-min intervals for 60 min according to the above method, and the corresponding degradation rates were calculated to plot the photocatalytic degradation curve.

## 4. Conclusions

In this study, Na_2_Ti_3_O_7_ nanobelts were successfully prepared via the hydrothermal method, and then, TiO_2_∙xH_2_O was obtained via ion exchange, which was followed by calcinating TiO_2_∙xH2O in a muffle furnace at 500 °C for 2 h to obtain TiO_2_(B) nanobelts. The belt-like morphology of Na_2_Ti_3_O_7_ was maintained throughout the preparation process, which made the TiO_2_(B) have good adsorption performance and enhanced the photocatalytic performance of the TiO_2_(B) photocatalyst. Using AgNO_3_ and Na_2_HPO_4_∙12H_2_O as reactants, Ag_3_PO_4_ nanoparticles were generated on the surface of the TiO_2_(B) nanobelts by a simple in situ method, thereby successfully preparing Ag_3_PO_4_/TiO_2_(B) heterojunction composites. The pure TiO_2_(B) nanobelt degraded the simulated pollutant RhB slowly, with a degradation rate lower than 20% after 30 min of degradation. By combining the TiO_2_(B) nanobelts with Ag_3_PO_4_, the performance of the composite photocatalyst was rapidly improved with the increase in the molar ratio of Ag_3_PO_4_, and the composite was able to degrade nearly 100% of RhB in 30 min when the molar ratio of Ag_3_PO_4_ to TiO_2_ was increased to 1.5:1. The improvement of the photocatalytic performance of the Ag_3_PO_4_/TiO_2_(B) composite was mainly attributed to the successful construction of heterojunctions between Ag_3_PO_4_ and TiO_2_(B), thereby greatly inhibiting the recombination of photogenerated electron–hole pairs and enhancing the light quantum yield of the photocatalyst. In addition, the introduction of Ag_3_PO_4_ in the composite successfully extended the light absorption range of the photocatalyst into the visible region, thus improving the utilization of simulated sunlight. Based on the above two reasons, the photocatalytic performance of the Ag_3_PO_4_/TiO_2_(B) composite was significantly improved.

## Figures and Tables

**Figure 1 molecules-26-06987-f001:**
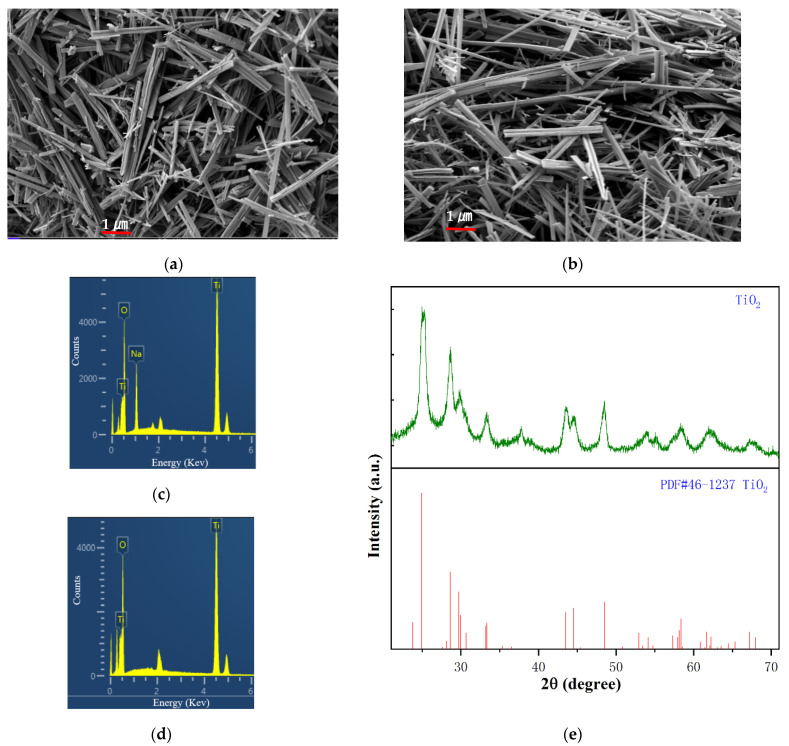
(**a**) SEM image of Na_2_TiO_7_ nanobelt; (**b**) SEM image of TiO_2_ nanobelt; (**c**,**d**) EDS spectra of TiO_2_ and Na_2_Ti_3_O_7_ nanobelt; (**e**) XRD patterns of TiO_2_. EDS, energy-dispersive X-ray spectroscopy.

**Figure 2 molecules-26-06987-f002:**
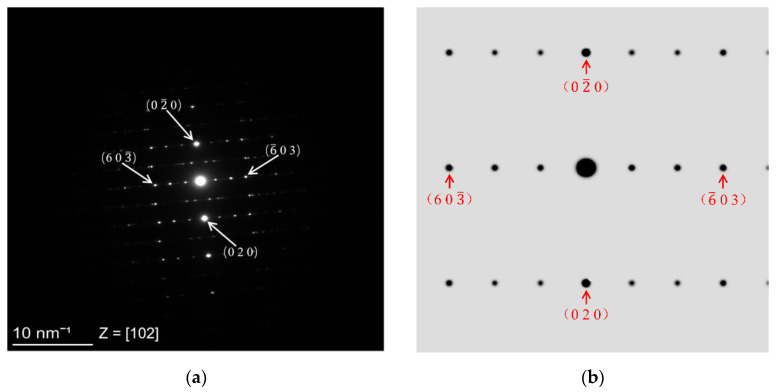
(**a**,**b**) Diffraction patterns of TiO_2_(B) nanobelt; (**c**,**d**) HRTEM (High-Resolution Transmission Electron Microscopy) images of TiO_2_(B) nanobelt.

**Figure 3 molecules-26-06987-f003:**
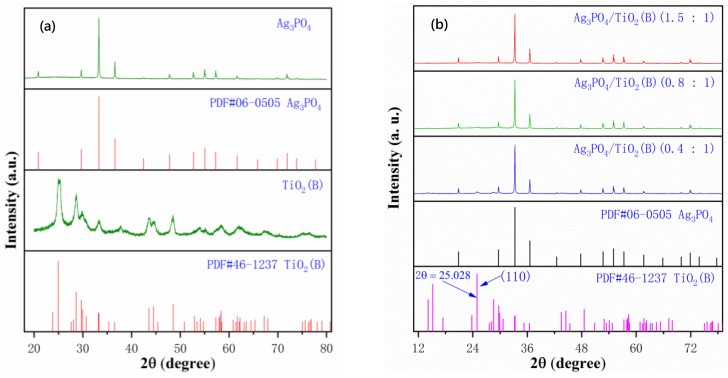
(**a**) XRD patterns of Ag_3_PO_4_ and TiO_2_(B); (**b**) XRD patterns of Ag_3_PO_4_/TiO_2_(B) with different molar ratios.

**Figure 4 molecules-26-06987-f004:**
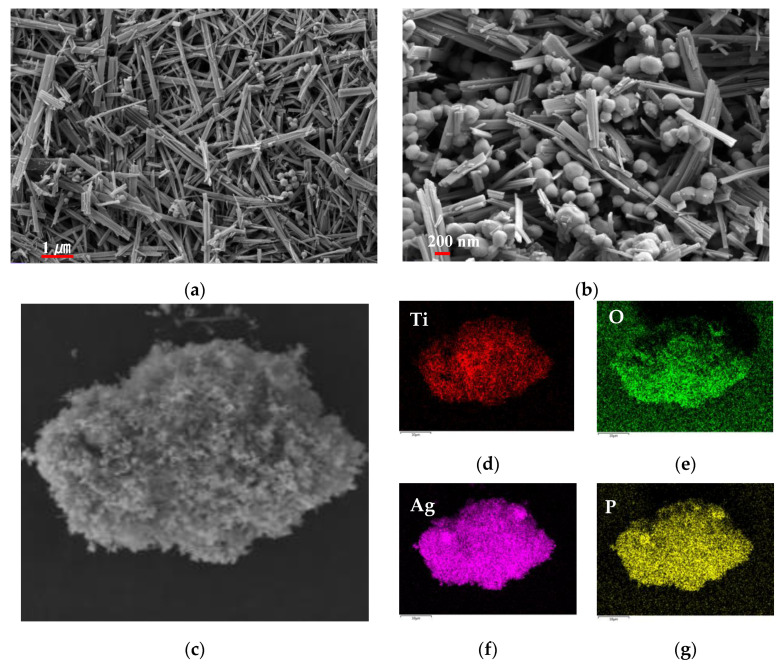
(**a**,**b**) SEM images of Ag_3_PO_4_/TiO_2_ composite; (**c**) The SEM image corresponding to map; (**d**) Map of Ti; (**e**) Map of O; (**f**) Map of Ag; (**g**) Map of P.

**Figure 5 molecules-26-06987-f005:**
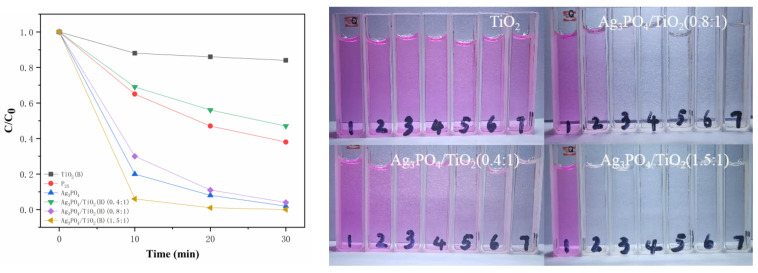
Photocatalytic degradation curves and images of Ag_3_PO_4_/TiO_2_ composite.

**Figure 6 molecules-26-06987-f006:**
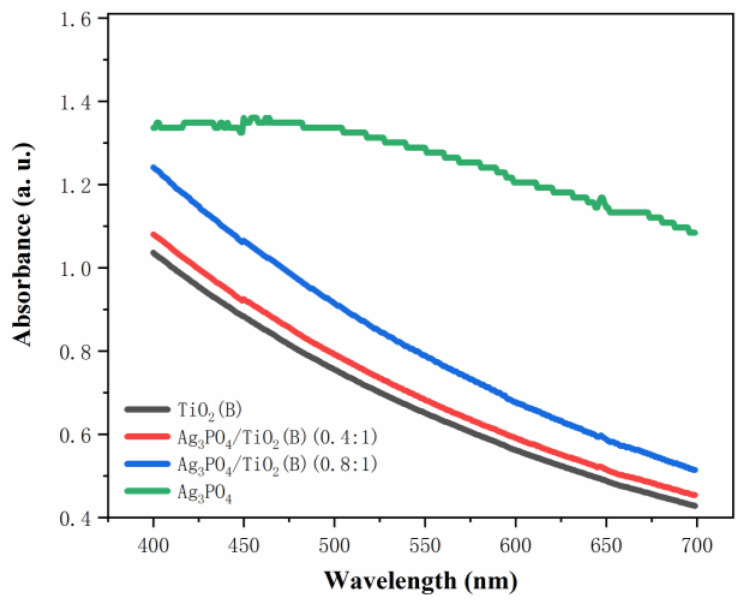
Continuous spectra of absorbance of synthesized samples.

**Figure 7 molecules-26-06987-f007:**
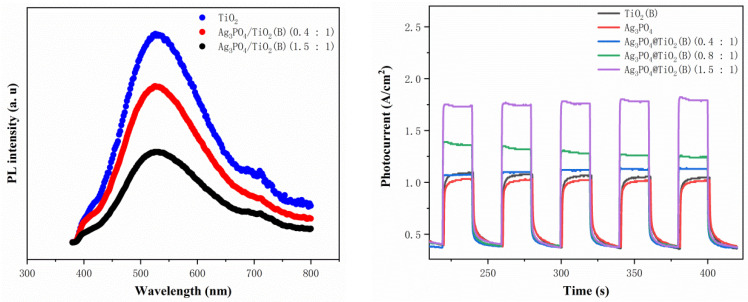
PL and photocurrent spectra of synthesized samples.

## Data Availability

Not applicable.

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
