# Peer review of "Preparation of Ag3PO4/TiO2(B) Heterojunction Nanobelt with Extended Light Response and Enhanced Photocatalytic Performance"

_molecules, 2021, doi:10.3390/molecules26226987_

Round 1

Reviewer 1 Report

Review

Journal: Molecules

Manuscript ID: molecules-1451770

Title: Preparation of Ag3PO4/TiO2 Heterojunction Nanobelt with Extended Light Response and Enhanced Photocatalytic Performance

The authors report the synthesis of TiO2 nanobelts and their modification with Ag3PO4 to increase their photocatalytic activity. Overall, the discussion is in good agreement with the acquired results, it can be easily followed, and it is well supported by the figures. The English of the paper is rather good, it is coherent, I only came across minor issues. However, conceptually, beyond the results presented, I have doubts. In my opinion, when it comes to the discussion of the results, significant improvements can be made. The paper can be accepted after major revisions if the issues raised have been addressed.

Minor comments:

  • A minor proofreading should be carried out. A few examples: an SEM image à a SEM image; ultrasonic shaking à ultrasonication; compositiion à There are some long sentences that could be reconstructed into several shorter ones, and some repetitions could also be avoided this way (e.g., common(ly found) in nature).
  • In the conclusions section instead of “sunlight” the authors should write “simulated sunlight”.
  • The figure caption of Fig. 1e should be corrected from “XRD spectra” to “XRD patterns”.
  • In Figs. 1c–d the x axes should be included.
  • In Figs. 1 and 4 the scale bars should be made more visible (like in Fig. 2).
  • In Fig. 3b, the diffraction peak at ~25 2θ° was misidentified. It does not correspond with the (110) plane of rutile. It corresponds with the (101) plane of anatase.
  • In Figs. 4c–g the corresponding elements should be made more visible.
  • In Fig. 6 the unit of dimension at the y axis is incorrect (Absorption (A)).
  • The abbreviation “TEM” should be used consistently (transmission electron microscopy and projection electron microscopy are used both).
  • It is increasingly accepted that using dyes to evaluate the photocatalytic activity of the samples is not the best approach. This is especially true for rhodamine B, because there are numerous papers in the literature that highlight the fact that rhodamine B can facilitate its own degradation by sensitizing the catalyst. Thus, it is difficult to differentiate the cause and proportions of the degraded dye. Did you consider this issue?

Major and crucial comments that are essential to be included and answered before accepting the paper:

  • I have found numerous papers regarding the modification of TiO2 nanobelts with Ag3PO4. Papers about using these materials i) in composites for dye degradation (1016/j.seppur.2021.119400, 10.1016/j.jcis.2015.11.072), ii) as heterostructures for dye degradation (10.1016/j.apsusc.2012.06.033), iii) under sunlight irradiation focusing on heterojunctions (10.1016/j.jpcs.2017.11.028), iv) and their in-depth characterization (10.1016/j.jssc.2021.122655) has already been published. In some of the cited papers the morphology is also the same. Because of this, I think it would be important for the authors to differentiate and highlight the novelty of their work better.
  • The authors made considerable efforts to ensure the formation of the nanobelt morphology. Starting from commercial TiO2 they transformed it to Na2Ti3O7 and transformed it back to TiO2. Naturally, these steps lower the economic feasibility of the synthesis procedure (and thus, the practical applicability). My biggest concern is that the authors have not included the importance of morphology neither in the introduction section nor during the discussion of the results. Why was it important to ensure the formation of this morphology? What was the reason for carrying out the synthesis procedure this way? What does the shape-tailored approach add to this research? This is an absolute must to explain and include in the manuscript in detail.
  • The authors used a high-performance lamp to irradiate their catalysts. Thus, it would be beneficial to include a commercial reference, such as P25 or another visible-light-active TiO2, to which the results should be compared. This would also make it possible to evaluate the change in photocatalytic activity that is caused by the unique morphology.
  • The authors highlight that the prepared titania was not anatase, rutile or brookite and that the crystal system is monoclinic. This piqued my interest. Did you know that titanium dioxide has other crystal phases? These include akaogiite and riesite. What is more, both of them are monoclinic! You can read a little about them in the following: https://www.sciencedirect.com/topics/earth-and-planetary-sciences/anatase. The authors should consider these rare crystal phases and investigate whether the material they produced is either of these crystal phases. This would greatly increase the value of the paper.

Reviewer 2 Report

The presented article is devoted to composite materials for photocatalysis with the Ag3PO4 / TiO2 heterojunction. This work is quite interesting, but there are some remarks and comments.

1) The monoclinic modification of titania like synthesized in this work is usually called beta-TiO2 or B-TiO2.

2) In the introduction does not describe the originality and novelty of the study. There is no information about existing works devoted to the Ag3PO4 / TiO2 system. These must be added to introductions.

3) There is no information about existing works devoted to the photocatalytic performances of monoclinic titania. These must be added to introductions.

4) Please use only “XRD pattern”. Not “XRD spectra” or “XRD diffraction patterns”.

5) When specifying the cell parameters, it is necessary to indicate the measurement errors.

6) TEM – Transmission Electron Microscopy, not Projection..

7) Fig 2. For TEM diffraction it is necessary to notice the zone. [102] for example.

8) for figure 4 add in the graph description the elements list. “d) – Ti map” for example.

9) Fig 5 and lines 218-219. degradation rate is not correct term because (A0-An)/A0 = 1-Cn/C0, where C is concentration. C/C0 vs time or ln(C/C0) vs time are more commonly presentation of photocatalytic data.

10) For all equipment units add the company and country of production.

Round 2

Reviewer 1 Report

The authors did a remarkable job answering my questions. I am very content with the answers. Due to the corrections, the scientific value has been greatly increased, and the novelty factor of the paper has been very well established. The paper can be accepted in its current form. I only have some minor comments left that could be useful to include before finalization:

  • There is a typo left in the manuscript: the word “compositiion” contains two “i” letters.
  • At the first occurrence, you should include what does “(B)” mean. If I am not mistaken, it is short for “bronze phase”?
  • Regarding the EDX spectra, I did not mean you to extend the x axis past the 6 keV (kilo-electronvolt) value, because it is clear that there is no valuable information after that point. Feel free to revert this change. What I actually meant, is that the title of the x axis is missing. So, what is missing is: Energy (keV)

Author Response

  • There is a typo left in the manuscript: the word “compositiion” contains two “i” letters.

Response:

Thank you for the comments. This error has been corrected in Line 79 of the revised manuscript.

  • At the first occurrence, you should include what does “(B)” mean. If I am not mistaken, it is short for “bronze phase”?

Response:

Thank you for the comments. This error has been corrected in Line 21 of the revised manuscript.

  • Regarding the EDX spectra, I did not mean you to extend the x axis past the 6 keV (kilo-electronvolt) value, because it is clear that there is no valuable information after that point. Feel free to revert this change. What I actually meant, is that the title of the x axis is missing. So, what is missing is: Energy (keV)

Response:

Thank you for the comments. This error has been corrected and fig. 1c-d in the revised manuscript has been replaced.